# Integrative Investigation of Root-Related mRNAs, lncRNAs and circRNAs of “Muscat Hamburg” (*Vitis vinifera* L.) Grapevine in Response to Root Restriction through Transcriptomic Analyses

**DOI:** 10.3390/genes13091547

**Published:** 2022-08-27

**Authors:** Jingjing Liu, Hui Li, Lipeng Zhang, Yue Song, Juan He, Wenping Xu, Chao Ma, Yi Ren, Huaifeng Liu

**Affiliations:** 1Department of Horticulture, College of Agriculture, Shihezi University, Shihezi 832003, China; 2Xinjiang Production and Construction Corps Key Laboratory of Special Fruits and Vegetables Cultivation Physiology and Germplasm Resources Utilization, Shihezi 832003, China; 3Department of Plant Science, School of Agriculture and Biology, Shanghai Jiao Tong University, Shanghai 200240, China

**Keywords:** grapevine, root restriction, root development, lncRNA, circRNA

## Abstract

Root restriction is a physical and ecological cultivation mode which restricts plant roots into a limited container to regulate vegetative and reproduction growth by reshaping root architecture. However, little is known about related molecular mechanisms. To uncover the root-related regulatory network of endogenous RNAs under root restriction cultivation (referred to RR), transcriptome-wide analyses of mRNAs, long non-coding RNAs (lncRNAs), and circular RNAs (circRNAs) involved in root development were performed. During root development, RR treatment had a positive effect on root weight, typically, young roots were significantly higher than conventional cultivation (referred to NR) treatment, suggesting that root architecture reconstruction under RR was attributed to the vigorous induction into lateral roots. Furthermore, a total of 26,588 mRNAs, 1971 lncRNAs, and 2615 circRNAs were identified in root of annual “Muscat Hamburg” grapevine by the transcriptomic analyses. The expression profile of mRNAs, lncRNAs and circRNA were further confirmed by the quantitative real-time PCR (RT-qPCR). Gene ontology enrichment analysis showed that a majority of the differentially expressed mRNAs, lncRNAs and circRNAs were enriched into the categories of cellular process, metabolic process, cell part, binding, and catalytic activity. In addition, the regulatory network of endogenous RNAs was then constructed by the prediction of lncRNA-miRNA-mRNA and circRNA-miRNA-mRNA network, implying that these RNAs play significant regulatory roles for root architecture shaping in response to root restriction. Our results, for the first time, the regulatory network of competitive endogenous RNAs (ceRNAs) functions of lncRNA and circRNA was integrated, and a basis for studying the potential functions of non-coding RNAs (ncRNAs) during root development of grapevine was provided.

## 1. Introduction

“Muscat Hamburg” (*Vitis vinifera* L.) is a middle and late ripening variety of European subspecies with strong rose fragrance, which is popular among consumers. It is widely planted worldwide and can be used as a variety for production of table grape, juice, and wine. To be inspired by potted practice, from the 1990s, agronomists turned to explore the cultivation ways of limiting roots of horticultural crops, named the root restriction cultivation (RR) [1]. RR refers to adjust the aboveground and underground part growth to re-coordinate the relationship between vegetative and reproduction growth by limiting the roots into a certain space [2]. It is supposed that RR effectively improves the balance of fruit yield and quality by reducing the redundant growth. In grapevine, RR has positive roles in sugar metabolism and anthocyanin-related gene expression, resulting in up-regulated accumulation of sugar and anthocyanin in fruits [3]. In general, the root architecture is obviously reshaped during RR which mainly significantly increases the number of lateral roots and improves the root regeneration rate [4]. In addition, RR can also accelerate the grapevine ripening [5], increases stem sucrose content [6], inhibits shoot growth [7] and root growth [8], reduces plant height, leaf area, and dry plant mass [9].

Transcriptomic analysis has shown that more than 90% DNA are transcribed and most of that are ncRNAs in plant genome [10,11]. NcRNAs refer to RNAs that lack the ability to encode proteins, which were initially regarded as inessential transcription “noise”. However, numerous research have demonstrated that ncRNAs have significant effect on various biological processes [12]. Generally, ncRNAs mainly consist of microRNAs (miRNAs), ncRNAs, and circRNAs. Studies have shown that some circRNAs and lncRNAs, harboring multiple binding sites for miRNAs, could regulate the activity of miRNAs by sponging miRNAs [13,14,15]. It is supposed that ncRNAs act as ceRNAs that sequester and suppress miRNA activity [16,17], and play vital roles in the development of plants and animals [18,19,20]. However, the sponge effect of lncRNAs and circRNAs for miRNAs has not been identified and verified in grapevine.

The ncRNAs with length over 200 nucleotides are defined as lncRNAs, which are always expressed at low level and have little conservation [21]. LncRNAs are found to be involved in post-transcriptional gene regulation and chromatin modifications [22,23]. With the rapid development of sequencing techniques, thousands of lncRNAs have been identified in several model plants, including Arabidopsis [24], maize [25], rice [26], poplar [27], and cotton [28]. CircRNAs are a novel type of ncRNAs formed by the back-splicing of one or more exons. CircRNAs occur widely in eukaryotic cells with tissue- and development-specificity [29]. CircRNAs are widely distributed in plants and have been identified in *Arabidopsis* [30], tomato [31], soybean [32], and rice [33] using deep RNA-seq and bioinformatic tools. In addition, a small number of circRNAs can be translated directly in mammals [34]. Since circRNAs plays an important regulatory role at the transcriptional or post-transcriptional level, and it is important to study the expression of circRNAs for a transcriptional regulatory network. However, it is unclear whether lncRNAs and circRNAs play a specific physiological role in grapevine. Although some research have focused on the roles of lncRNAs and circRNAs in plants, lncRNAs and circRNAs involved in root development or root architecture reconstruction during RR have not been identified in grapevine.

In this study, ceRNA networks of lncRNAs, circRNAs, and mRNAs were integrated based on high-throughput sequencing data. For the first time, the potential ceRNA functions of lncRNAs and circRNAs involved in root growth and development have been investigated. The results provided a basis for deciphering the regulatory networks of ncRNAs related to root development and architecture reshaping during root-restricted cultivation.

## 2. Results

### 2.1. Root Phenotype under Conventional and Root-Restricted Cultivation

To observe differences of root phenotype, one-year-old self-rooted grapevine cv. “Muscat Hamburg” was planted by NR and RR, respectively. The entire root system was sampled and photographed to record root phenotype during different developmental stages (Figure 1). A total of twelve sampling time points were conducted (at one development stage, root systems were simultaneously collected from both cultivation methods). The phenotype of root samples from twelve different times of NR and RR were marked as NR1-12 and RR1-12 (Figure 1), respectively. According to the results, similar root formation orders including new adventitious roots, absorbing roots and secondary lateral roots have been observed between NR and RR. Finally, the old roots degenerated, and young roots developed into the main root system. However, from the seventh sample (NR7 and RR7), the root morphology was significantly changed visually between NR and RR cultivation during the developmental process. Before the seventh sampling, new adventitious roots were generated and elongated continuously under NR and RR. After the seventh sampling, the old roots gradually aged and disappeared from bottom to top. The adventitious roots of NR became coarse and gradually lignified to brown, and less adventitious roots were produced. However, the adventitious roots under RR further elongated to brown, differentiated more lateral roots, and produced a great deal of adventitious roots continuously.

In order to describe the root architecture difference of cv. “Muscat Hamburg” in response to root restriction, the weight of young roots, old roots, and total roots were further obtained (Figure 2). The weight of all three kinds of roots increased with developmental process. In the later stage of development, the weight of total roots was significantly different at the first, sixth, seventh, ninth, tenth, eleventh, and twelfth time points between NR and RR, and RR9–12 roots were heavier than NR9–12. The weight of old roots with different treatments was significant at the first, sixth, seventh, eighth, ninth, and eleventh time points. NR1 roots were lighter than RR1, and NR6–9 roots were heavier than RR6–9. However, the weight of young roots was largely derived from the lateral root morphogenesis, and RR4–5 and RR7–12 roots were significantly heavier than NR4–5 and NR7–12. It showed that the difference in total roots weight were caused by enhancing the induction of young lateral roots, which can also be seen in Figure 1 for the difference in young lateral roots. The decrease in the old root weight of NR11 in Figure 2b might be caused by sampling errors or individual differences in plants. Combining the above results, it is proved that the root weight increasing and architecture remodeling of grapevine changed under RR conditions, mainly due to the increase of lateral roots.

### 2.2. Sequencing Statistics in Different Roots Samples

Before the seventh sampling, although NR generated fewer adventitious roots than RR, new adventitious roots of NR were continuously generated and elongated. After the seventh sampling, NR lateral roots were gradually lignified and adventitious roots became less. Therefore, seventh time point under NR may be the inflection point for root development. We considered two comparison modes, one is the comparison of root development under conventional cultivation (NR7 and NR12), and the other group is used to compare the differences of root development between NR and RR. Therefore, the time point with the greatest difference was selected in RR. Over the whole stage, more lateral roots were differentiated under the RR, and a large number of adventitious roots were continuously produced. However, at twelfth time point in RR, the weight of young roots was largely different, which was in accordance with the phenotype of the root system. To systematically investigate the potential regulatory roles of genes and ncRNAs involved in root development in cv. “Muscat Hamburg”, we first collected the roots derived from two different stages of development (NR7 and NR12) in NR and roots derived from the stage with abundant lateral root in RR (RR12) and performed integrative transcriptomic analyses.

As a result, 59.62 G CleanReads from a total of 9 samples (NR7_1, NR7_2, NR7_3, NR12_1, NR12_2, NR12_3, RR12_1, RR12_2 and RR12_3) were obtained. The number of CleanBases ranged from 5.35 to 8.69 G in each sample, the Q30 base distribution ranged from 93.56% to 94.28%, and the average GC content was 47.07% (Table 1). The statistical assessment confirmed that the dataset was highly reliable.

Based on the expression pattern of mRNAs, lncRNAs, and circRNAs, correlation and principal component analysis (PCA) of the nine samples were conducted. The results indicated that the three independent biological replicates were repeatable (Figure 3a–c). Meanwhile, the nine samples were divided into three clusters (NR7, NR12, and RR12), and PCA showed that the contribution rates of the first principal component (PC1) of mRNA, lncRNA, and circRNA were 59.55, 28.30, and 16.85%, respectively. The closer the clustering distance or PCA distance of the samples, the more similar the samples were (Figure 3d–f).

### 2.3. Identification and Characterization of lncRNAs and circRNAs

There were different distributions of lncRNAs and circRNAs on each chromosome. The chromosome distribution analysis showed that most of the circRNAs and lncRNAs were generated from chromosome 18 and chromosome 8 (Figure 4a). The length of lncRNAs and circRNAs in grapevine were different, and the distribution of most lncRNAs and circRNAs were regular (Figure 4b). The length of 612 circRNAs (23.40%) and 149 lncRNAs (7.56%) were greater than 2000 nt, and 350 circRNAs were less than or equal to 200 nt. The number of lncRNAs and circRNAs in the range of 201–2000 nt decreased with their sequence length (Figure 4b). Based on the position relationship between lncRNA and known protein coding transcripts, FEELnc software was used to count lncRNA types according to three levels, including direction, type, and location. A total of 657 antisense lncRNAs and 1308 sense lncRNAs were identified (Figure 4c). Bedtools software was used to classify the position relationship between circRNA and the coding genes, and five types of circRNAs were found; the largest number of circRNA of the sense-overlapping type contains 1878 circRNAs (72%). Intronic circRNAs was the lowest proportion containing 35 circRNAs (1.34%). Moreover, the percentage of exonic-derived circRNAs, antisense-derived circRNAs, and intergenic-derived circRNAs accounted for 15.64% (409 circRNAs), 7.88% (206 circRNAs), and 3.33% (87 circRNAs), respectively (Figure 4d).

### 2.4. Differential Expression Analyses of mRNA, lncRNAs, and circRNAs

There were three comparison groups (NR12_vs_NR7, RR12_vs_NR7, RR12_vs_NR12) used to analyze the differential expression of mRNA, lncRNAs, and circRNAs. A total of 26,588 mRNAs were detected in all groups. The numbers of differentially expressed genes (DEGs) were 2320, 1864, and 2440 for NR12_vs_NR7, RR12_vs_NR7, and RR12_vs_NR12, respectively. A total of 2615 circRNAs were predicted. However, a few circRNAs were differentially accumulated in three comparison groups, which were 16, 17, and 9, respectively. Additionally, 1971 lncRNAs were identified, and 176, 173, and 137 were differentially expressed in NR12_vs_NR7, RR12_vs_NR7, and RR12_vs_NR12, respectively (Table 2).

We compared the mRNAs, lncRNAs, and circRNAs expression profiles in the three comparison groups. We found 170 mRNAs, 5 lncRNAs, and 0 circRNA that were shared by three comparison groups (Figure 5a–c). There were 2320 mRNAs (1329 up-regulated and 991 down-regulated), 176 lncRNAs (110 up-regulated and 66 down-regulated), and 16 circRNAs (9 up-regulated and 7 down-regulated) that were differentially expressed in NR12_vs_NR7. About 2440 mRNAs (1237 up-regulated and 1203 down-regulated), 173 lncRNAs (84 up-regulated and 89 down-regulated), and 17 circRNAs (9 up-regulated and 8 down-regulated) were differentially expressed in RR12_vs_NR12. Moreover, there were 1864 mRNAs (1222 up-regulated and 642 down-regulated), 137 lncRNAs (79 up-regulated and 58 down-regulated), and 9 circRNAs (including 7 up-regulated and 2 down-regulated) that were differentially expressed in RR12_vs_NR7 (Figure 5d–f).

### 2.5. Functional Enrichment Analyses

To explore the functions of the mRNAs, Gene Ontology (GO) and Kyoto Encyclopedia of Genes and Genomes (KEGG) analyses were performed. The GO terms of DEGs mainly related to biological regulation, metabolic process, cellular process, cell, cell part, binding, catalytic activity, and transporter activity in three comparison groups (NR12_vs_NR7, RR12_vs_NR7, RR12_vs_NR12). KEGG pathway analysis was also performed to further explore the functions of mRNAs. The results showed that biological functions of mRNAs were significantly enriched in three comparison groups, including lipid metabolism, carbohydrate metabolism, biosynthesis of other secondary metabolites, and signal transduction (Figure 6).

To investigate the potential function implications of lncRNAs, after obtaining differentially expressed lncRNAs, GO and KEGG enrichment analyses were also performed based on the functions of neighboring genes. The results showed that these lncRNAs were enriched in the cellular process, metabolic process, cell, cell part, binding, and catalytic activity in three comparison groups. The KEGG enrichment analysis showed that these lncRNAs were clustered into nucleotide metabolism, carbohydrate metabolism, and signal transduction in three comparison groups (Figure 7).

To further understand the potential functions of circRNAs, GO and KEGG analyses of the host genes of differentially expressed circRNAs were performed. Based on GO annotation, most the host genes of circRNAs in the root were annotated to the cellular process, metabolic process, cell, cell part, binding, and catalytic activity in three comparison groups. The parent genes of circRNAs were involved in 5 KEGG pathways and significantly enriched in metabolism of other amino acids, lipid metabolism, global and overview maps, energy metabolism, and amino acid metabolism (Figure 8).

### 2.6. Experimental Validation of the circRNA Candidates

In the present study, we experimentally tested and verified the circRNA predictions in the grapevine. Since the majority of circRNA were derived from back-splicing of mRNA transcripts, convergent primers and divergent primers were designed and used to amplify the linear transcript and back-splicing sites using cDNA and genomic DNA (gDNA) as template, respectively. Unlike convergent primers, divergent primers could amplify the back-splicing junctions in cDNAs synthesized by the random primers but not in gDNA. Moreover, we successfully validated the circRNA, circrRNA_2377, potentially involved in root development. The PCR amplification products were further analyzed by agarose gel electrophoresis and Sanger sequencing to confirm the occurrence of back-splicing (Figure 9). There was some amplification in gDNA using divergent primers due to the production of primer dimers.

### 2.7. Real-Time Quantitative PCR (RT-qPCR)

To verify the reliability of the sequencing data profiles, RT-qPCR was used to analyze the candidate genes related to root development (Figure 10). We randomly selected one mRNA (VIT_06s0061g00310), two lncRNAs (TCONS_00012993 and TCONS_00037367) and one circRNA (circRNA_2377) involved in the regulation of root development. The expression patterns of VIT_06s0061g00310, TCONS_00012993, TCONS_00037367 in the three comparison groups and circRNA_2377 in the two comparison groups (RR12_vs_NR7 and RR12_vs_NR12) were consistent with the high-throughput sequencing results. However, circRNA_2377 in NR12_vs_NR7 was not consistent with the high-throughput sequencing, which might be due to its low expression levels (Figure 10).

### 2.8. CeRNA Network Analyses

It was supposed that lncRNA and circRNA generally contain one or multiple miRNA binding sites, and the method of miRNA target gene prediction can be used to identify lncRNA and circRNA which bind to miRNA. The functions of lncRNA and circRNA were clarified according to the functional annotations of miRNA target genes. In the present study, all the candidate ceRNA pairs (circRNA/lncRNA sequesters miRNA and miRNA targets mRNA) were first identified. Top 5 miRNA-circRNA interaction combinations were extracted based on minimum free energy according to RNAInter prediction. LncTar software was used to predict lncRNA-miRNA interactions. Furthermore, 3 miRNA-mRNA networks with a high degree of confidence were also predicted and extracted. Finally, the lncRNA/circRNA-miRNA-mRNA target interaction network diagram was obtained and plotted by Cytoscape software, and a hub prognostic ceRNA network was constructed consisting of 21 mRNAs, 14 miRNAs, 2 lncRNAs, and 1 circRNA (Figure 11).

As showed in Figure 11, a large proportion of mRNAs communicated with individual lncRNA and circRNA. LncRNA and circRNA acted as ceRNAs to communicate with multiple mRNAs by competing for specific miRNAs. These results suggested that the expression of lncRNA and circRNA potentially regulated gene expression through miRNA-mediated lncRNA/circRNA-mRNA ceRNA interactions, implying that ceRNA might be important for lncRNA and circRNA function during root development.

## 3. Discussion

Root restriction which through limiting the roots into a fixed space to re-coordinate the relationship between vegetative and reproductive growth, has been widely practiced in fruit production [1]. In this study, one-year-old cv. “Muscat Hamburg” was used as material to explore the regulatory mechanism involved in root architecture construction during conventional and root-restricted cultivation. We first observed that root morphogenesis was obviously different on the condition of NR and RR, in which lateral root formation was much more vigorous during the late development stage of RR (Figure 1). Previous research suggested that the root architecture was updated after RR [4], which is consistent with this study. During root development, RR treatment had a positive effect on root weight, typically, the young roots observed on the RR condition were heavier than NR treatment. The weight of young roots was largely derived from the lateral root morphogenesis. Therefore, root architecture reconstruction was attributed to the vigorous induction of lateral roots in response to RR condition.

The ceRNA hypothesis has been widely accepted since it was first reported in several years ago [18]. CeRNA theory has been well applied to understand the mechanism of human disease [35]; however, only a few researches have been carried out in plants [36]. The mRNAs can be directly transcribed into proteins, while lncRNAs and circRNAs can indirectly influence mRNAs expression by competitively binding to common miRNAs [18]. A few of ncRNAs have been found to exhibit different expression profiles in plant roots. However, the functions of these ncRNAs utilized to regulate root development have not been well characterized and deciphered. Therefore, we presented ceRNA network analysis to elucidate potential regulatory mechanisms during grapevine root development.

Recently, increasing studies indicate that ncRNAs play an important role during plant development. To better understand ncRNAs expression and underlying regulatory mechanisms in grapevine roots, in this study, a total of 26,588 mRNAs, 1971 lncRNAs, and 2615 circRNAs were identified through transcriptomic analyses in grapevine root, implying that they are abundant in developing roots. At the same time, these genes were found to be implicated in various biological processes, including catalytic activity, nucleotide metabolism, carbohydrate metabolism, signal transduction, lipid metabolism, energy metabolism, and amino acid metabolism (Figure 6, Figure 7 and Figure 8). To verify the reliability of the sequencing data profiles, the candidate genes (VIT_06s0061g00310, TCONS_00012993, TCONS_00037367 and circRNA_2377) were analyzed by RT-qPCR (Figure 10).

A ceRNA hypothesis has been proposed that the lncRNAs, circRNAs, and mRNAs can act as ceRNAs that competitively bind to miRNAs, thereby regulating a wide range of biological and developmental processes [37,38,39,40,41,42]. In addition, circRNAs possibly act as another type of ceRNA to sequester miRNAs and suppress their activity. CircRNAs have been shown to contain multiple binding sites for miRNAs in animals. One of the well-studied examples is Cdr1as/ciRS-7 in human, which can bind miR-7 as a miRNA sponge and affect miR-7 target gene expression [43]. Based on the ceRNA hypothesis and the ceRNA network constructed in this study, a model of action for lncRNAs, circRNAs, miRNAs, mRNAs in response to the root development has been provided.

As an important key regulator, miRNAs have been reported that they are involved in root development [44,45,46,47,48,49,50,51,52]. For example, miR160 has been reported to be involved in root cap and root elongation formation [44]. MiRNAs target auxin response factors to influence the development of adventitious root [45]. MiRNAs are likely to be an RNA bridge between non-coding RNAs and mRNAs [46]. MiR166 reduced the number of lateral roots and symbiotic nodules, and induced ectopic development of vascular bundles in transgenic roots [47]. Research has shown that the miR169defg isoforms promote primary root growth and inhibit lateral root initiation [48]. MiR2111 is not only the critical shoot-to-root factor that positively regulates root nodule development, but also shapes root system architecture [49]. In addition, *IREH1* encoded a transcription factor that may be involved in root hair elongation and was targeted by miR3623-3P [50]. *SPL10*, one of the SPL family genes that represses lateral root growth in *Arabidopsis thaliana*, was significantly down-regulated in miR156 overexpressed plants, suggesting that miR156 play an important role in plant growth and development [51]. Meanwhile, miRNA3634-3p has been predicted to target *NAC22* (Figure 11), while *NAC1* gene has also been reported to promote lateral root development and its production activates the expression of two downstream auxin-responsive genes, *DBP* and *AIR3* [52]. Interestingly, in previous research, it was reported that vvi-miR3623-3p, vvi-miR3634-3p, and vvi-miR3640-3p were down-regulated with the growth of root development in conventional cultivation. Compared with conventional cultivation, vvi-miR156e, vvi-miR166a, vvi-miR2111-5p, vvi-miR482, and vvi-miR477a were down-regulated under root-restricted cultivation [4]. These miRNAs also appeared in ceRNA network (Figure 11). In this study, TCONS_00012993 and *RPS2* were up-regulated, they may be involved in the process of root development, which deserves further study. Based on the ceRNA hypothesis, lncRNAs and circRNAs binding miRNAs might play a critical role in the regulation of root growth, there is still a lot of work to be done to clarify how it plays the regulatory roles.

## 4. Materials and Methods

### 4.1. Plant Material and Treatments

The grapevine cuttings were the materials used in this study. One-year-old self-rooted cv. “Muscat Hamburg” (*V. vinifera* L.) were planted in greenhouse of the Fruit Tree Laboratory in Shanghai Jiao Tong University. In this study, the materials were divided into conventional cultivation (NR) and root-restricted cultivation (RR). In RR, 100 cuttings were planted in a container with diameter of 30 cm and height of 30 cm in the root zone (with holes around it), which were separated from the ground by a tray. The planting substrate was a 1:1:1 mixture of soil, organic fertilizer, and perlite. In the control culture, 100 plants with consistent physiological state were cultivated on the same substrate with a planting distance of 60 × 60 cm. Sampling was performed in a zigzag pattern to ensure that there was sufficient space for the root system developing. The treatment of the above-ground management was the same, and the growth of them was maintained without topping. The secondary shoots were trimmed and watered every 7 days. Moreover, the roots began to sample on 10 March 2021. The root samples of different cultivation models were taken in every 10 days intervals during the first 7 time points and at 15-day intervals at the last 5 time points. Finally, grapevine root samples from two different cultivation models at 12 time points (53, 63, 73, 83, 93, 103, 113, 128, 143, 158, 173, and 188 days after planting) were collected. At each sampling time point, 4–6 trees were selected as one biological replicates. All root samples were collected from control and treated plants, frozen in liquid nitrogen, and stored at −80 °C.

### 4.2. Library Construction and Illumina Sequencing

According to the root phenotype, samples from the seventh and twelfth under NR and the twelfth under RR were selected for sequencing, respectively. Three biological replicates were performed for each sample. Total RNA was extracted by the mirVana miRNA Isolation Kit (Ambion, TX, USA) following the manufacturer’s protocol. RNA integrity was evaluated using the Agilent 2100 Bioanalyzer (Agilent Technologies, Santa Clara, CA, USA). The samples with RNA Integrity Number (RIN) ≥ 7 were subjected to the subsequent analysis. The libraries were constructed using TruSeq Stranded Total RNA with Ribo-Zero Gold according to the manufacturer’s instructions. Then these libraries were sequenced on the Illumina sequencing platform (HiSeqTM 2500 or other platform) and 150 bp/125 bp paired-end reads were generated. Three biological replicates were analyzed. The samples were named as NR7_1, NR7_2, NR7_3; NR12_1, NR12_2, NR12_3; RR12_1, RR12_2, RR12_3, respectively.

### 4.3. Data Preprocessing and Genomic Alignment

Raw reads generated during high-throughput sequencing were fastq format sequences. In order to get high-quality reads that could be used for later analysis, raw reads needed to be further quality filtered. Trimmomatic software [53] was first used for adapter removing, and then low-quality bases and N-bases or low-quality reads were filtered out. Finally, we got high-quality clean reads. Using hisat2 [54] to align clean reads to the reference genome of the experimental specie (ftp://ftp.ensemblgenomes.org/pub/plants/release-48/fasta/vitis_vinifera/dna/Vitis_vinifera.12X.dna.toplevel.fa.gz accessed on 20 October 2021), the sample was assessed by genomic and gene alignment.

### 4.4. Transcript Splicing, lncRNA Prediction, and Gene Quantification

The result of alignment with the reference genome was stored in a binary file, called a bam file. Using the Stringtie [55] software to assemble the reads, the new transcript was spliced. Then the candidate lncRNA transcripts were selected by comparing the gene annotation information of the reference sequence produced by Cuffcompare [56] software. Finally, transcripts with coding potential were screened out by CPC [57], Pfam [58], and PLEK [59] to obtain lncRNA predicted sequences. The sequencing reads of each sample were aligned with the mRNA transcript sequences, known lncRNA sequences, and lncRNA prediction sequences by bowtie2 [60]. eXpress [61] was used for gene quantitative analysis, and the Fragments Per Kilobase per Million (referred to FPKM) value and counts value (the number of reads for each gene in each sample) were obtained.

### 4.5. Differential Expression Analysis and Functional Analysis

The estimateSizeFactors function of the DESeq [62] R package was used to normalize the counts, and the nbinomTest function was used to calculate *p*-value and foldchange values for the difference comparison. Differential transcripts with *p*-values ≤ 0.05 and foldchange ≥ 2 were selected to analyze GO and KEGG [63] enrichment of differential mRNA, lncRNA, and circRNA by Hypergeometric Distribution Test.

### 4.6. CircRNA Prediction, Expression Analysis and Interaction Research

In order to generate SAM file, we used BWA [64] software to align the sequencing reads of each sample with reference genome. Then CIRI [65] software was used to scan for PCC signals (paired chiastic clipping signals), and circRNA sequences were predicted based on junction reads and GT-AG cleavage signals. The RPM algorithm was used to quantify the circRNAs and to normalize the number of junctional reads counts. The foldchange was evaluated by DESeq. Enrichment of differentially expressed circRNAs was analyzed through the annotation information of circRNA source transcripts. PsRNATarget [66], RNAInter [67] and LncTar [68] software were used to predict lncRNA/circRNA-miRNA-mRNA interactions. Grapevine miRNAs were retrieved from miRBase database [69]. The lncRNA/circRNA-miRNA-mRNA target interaction network diagram was obtained and plotted by Cytoscape software [70].

### 4.7. Validation of circRNA in Grapevine

To confirm the grapevine circRNA_2377 that were predicted by the software, the convergent and divergent primers were designed using the Primer-Blast tools in NCBI (www.ncbi.nlm.nih.gov accessed on 20 October 2021) (Appendix A). Genomic DNA (gDNA) and total RNA were used as templates for the PCR validation of circRNAs. 2×Taq Master Mix (Vazyme, Nanjing, China) was used to detect circRNA by PCR amplification based on the cDNA and gDNA templates. The PCR procedure was as follows: 94 °C for 3 min; 39 cycles at 94 °C for 30 s, 55 °C for 15 s and 72 °C for 30 s; and then 1 cycle at 72 °C for 5 min. The PCR products were separated using agarose gel electrophoresis and then purified with a Vazyme FastPure^®^ Gel DNA Extraction Kit (v4.0). Sanger sequencing of the direct PCR products was performed to further verify the existence of the back-splicing site of the circRNAs predicted in grapevine.

### 4.8. RT-qPCR Analysis

The total RNA was isolated from grape roots using a modified CTAB method [3] and then transcribed to cDNA, which were then used as templates for RT-qPCR. To verify the mRNA, lncRNA, and circRNA identified by high-throughput sequencing, one mRNA (VIT_06s0061g00310), two lncRNAs (TCONS_00012993 and TCONS_00037367), and one circRNA (circRNA_2377) were selected for RT-qPCR validation (Appendix A). The RT-qPCR was performed on a CFX Connect Real-Time Detection System (Bio Rad, Hercules, CA, USA). The program settings are as follows: 95 °C for 150 s, followed by 40 cycles of 95 °C for 5 s and 60 °C for 30 s. The relative expression levels were analyzed by the 2^−ΔΔC^^T^ method [71]. Actin was used as an internal control for the mRNA, lncRNA, and circRNA (Appendix A). All RT-qPCRs were performed in triplicate.

## 5. Conclusions

Root architecture reconstruction under root restriction was attributed to the vigorous induction of lateral roots. Root restriction altered the expression profiles of mRNA, lncRNA, and circRNAs in grapevine. We found that two lncRNAs and one circRNA were potentially involved in root development. They might act as miRNA sponges for trapping miRNAs from its target genes via the ceRNA network, thus affecting root development. To our knowledge, this is the first report on integrating the regulatory network of ceRNAs functions of lncRNA and circRNA, and provided a basis for studying the potential functions of ncRNAs during root development of grapevine.

## Figures and Tables

**Figure 1 genes-13-01547-f001:**
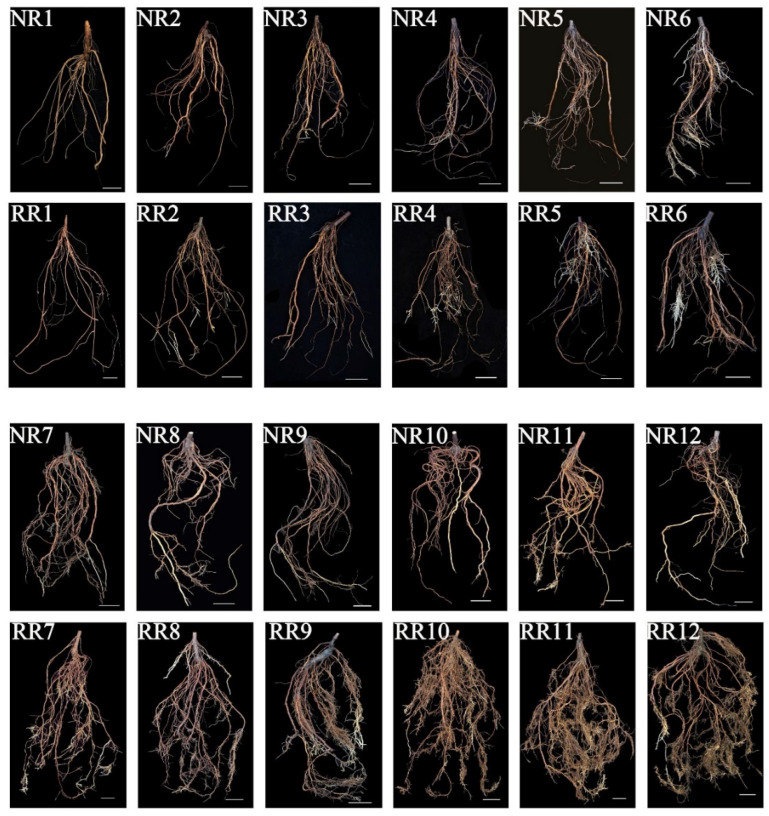
The developmental root system morphology of one-year-old self-rooted grapevine cv. “Muscat Hamburg” under NR and RR. Twelve different sampling points were recorded as NR1–12 and RR1–12, respectively. Scale bar = 5 cm.

**Figure 2 genes-13-01547-f002:**
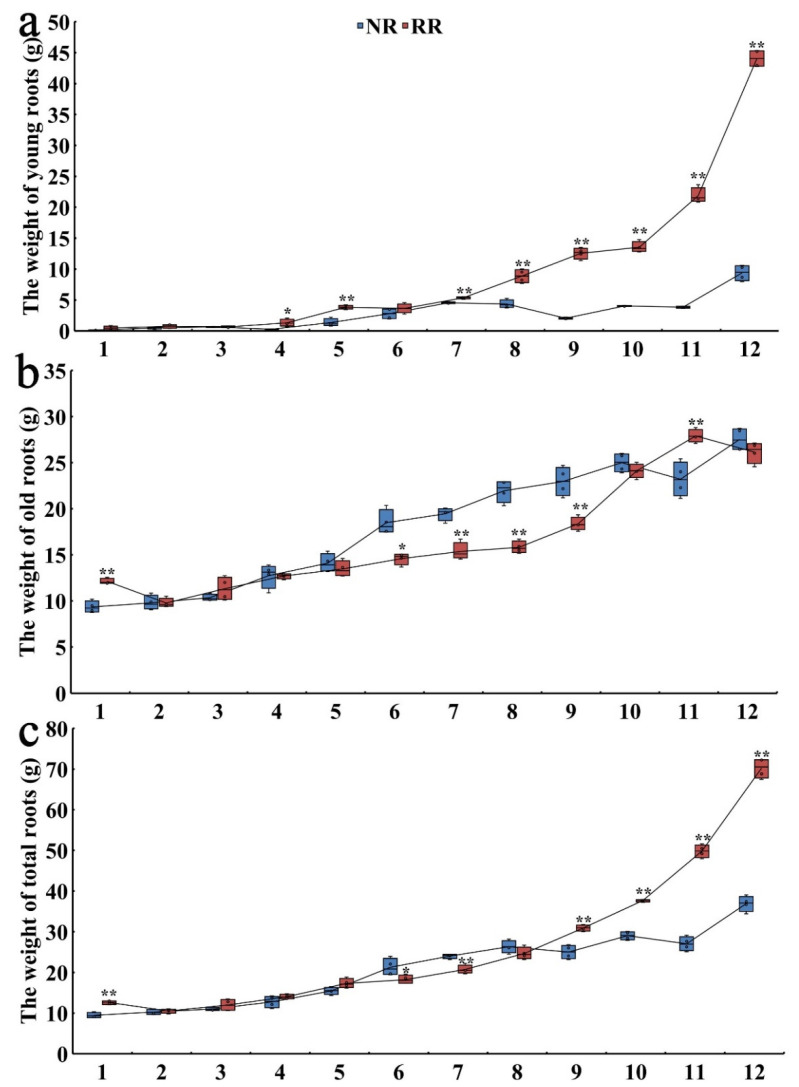
Root weight variations of one-year-old self-rooted grapevine cv. “Muscat Hamburg” on the condition of NR and RR. The weight of young roots (**a**), old roots (**b**), and total roots (**c**) was counted in NR and RR cultivation. 1–12 indicate the sample collection time points. Error bars show the standard error between four biological replicates. Values are statistically significant from the control cultivation based on *t*-test (* *p* < 0.05 and ** *p* < 0.01).

**Figure 3 genes-13-01547-f003:**
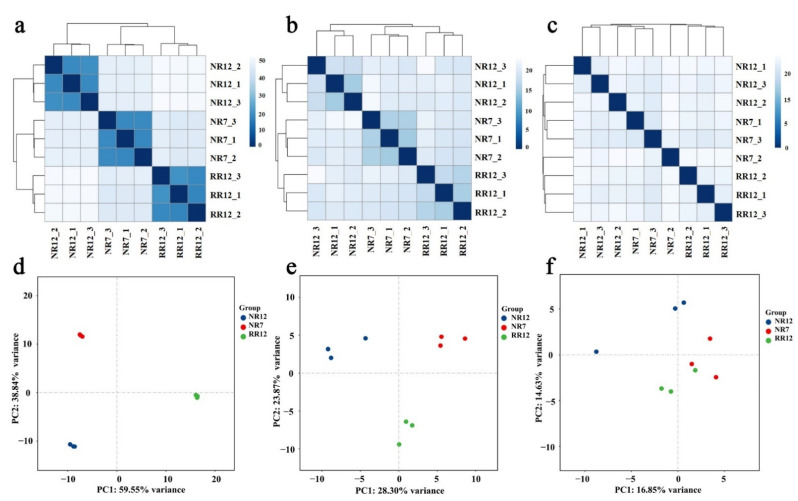
Correlation and PCA analyses of different samples. The sample-to-sample correlation analyses in mRNA (**a**), lncRNA (**b**), and circRNA (**c**). The horizontal axis and the vertical axis represent the corresponding sample names, and the color represents the size of the correlation coefficient. The PCA analyses among mRNA (**d**), lncRNA (**e**), and circRNA (**f**).

**Figure 4 genes-13-01547-f004:**
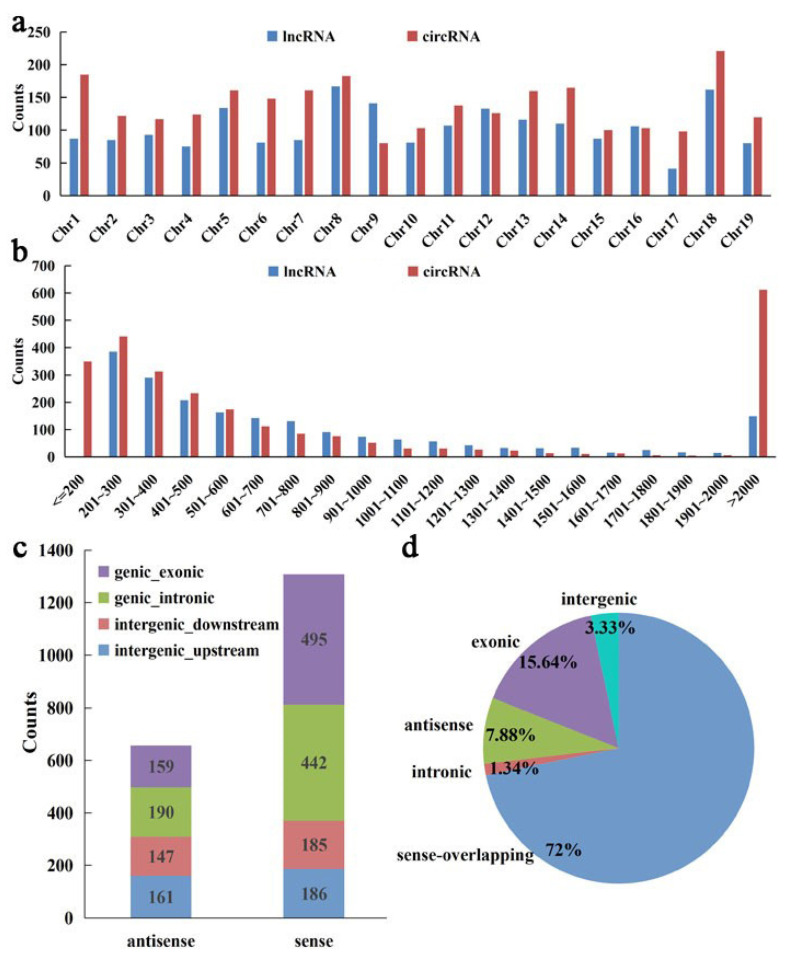
Comparison of the features of lncRNAs and circRNAs in the grapevine root. Chromosomes distribution (**a**), length distribution (**b**), and classification of lncRNAs (**c**) and circRNAs (**d**).

**Figure 5 genes-13-01547-f005:**
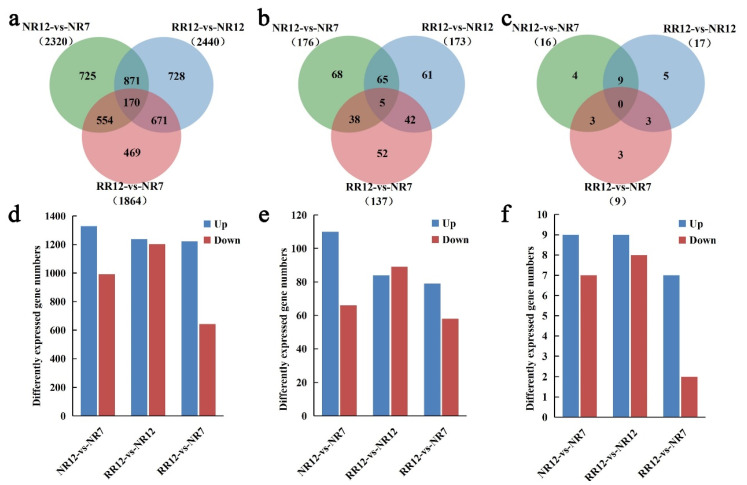
Comparative analysis of mRNAs, lncRNAs, and circRNAs. The specific mRNAs (**a**), lncRNAs (**b**), and circRNAs (**c**) shared by three comparison groups (NR12_vs_NR7, RR12_vs_NR7, RR12_vs_NR12). The mRNAs (**d**), lncRNAs (**e**), and circRNAs (**f**) differentially expressed in various sample groups.

**Figure 6 genes-13-01547-f006:**
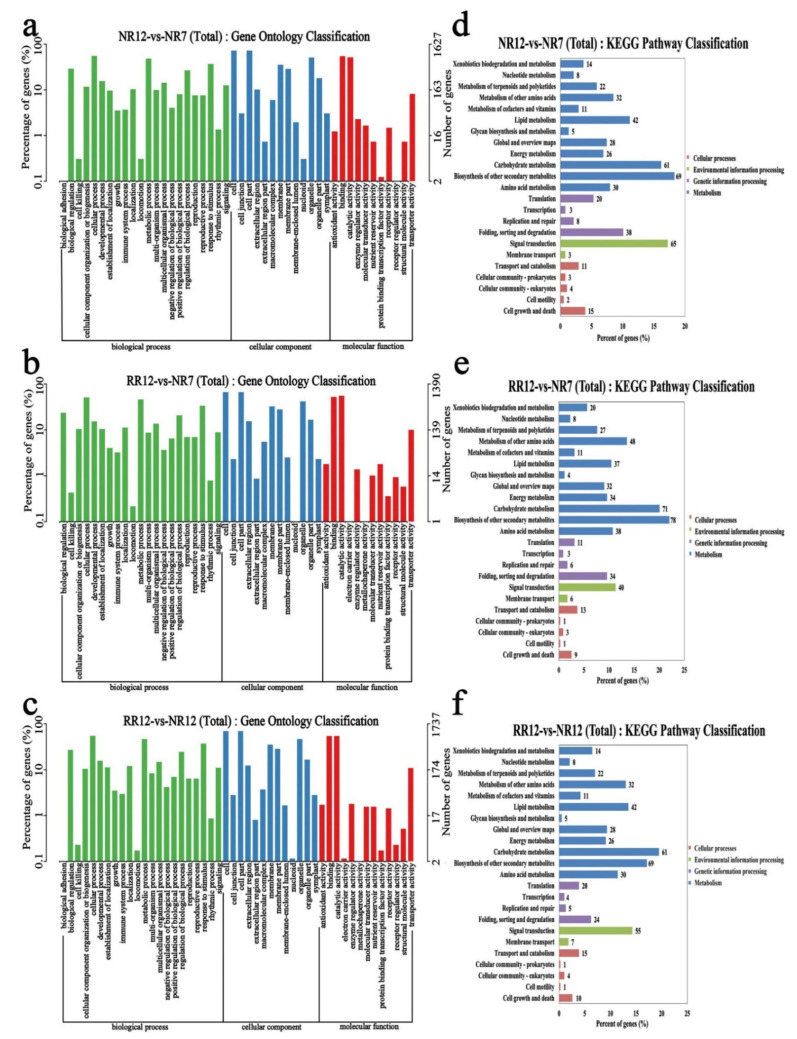
The functional analysis results of differentially expressed mRNAs. GO enrichment (**a**–**c**) and KEGG pathway (**d**–**f**) in NR12_vs_NR7, RR12_vs_NR7, RR12_vs_NR12.

**Figure 7 genes-13-01547-f007:**
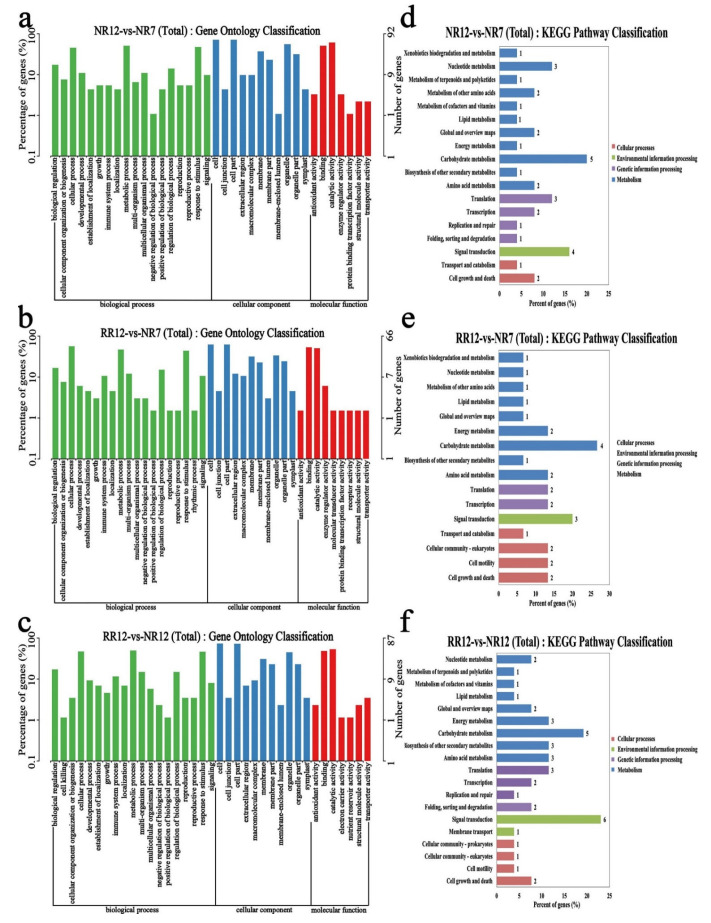
The functional analysis results of neighboring genes of differentially expressed lncRNAs. GO enrichment (**a**–**c**) and KEGG pathway (**d**–**f**) in NR12_vs_NR7, RR12_vs_NR7, RR12_vs_NR12.

**Figure 8 genes-13-01547-f008:**
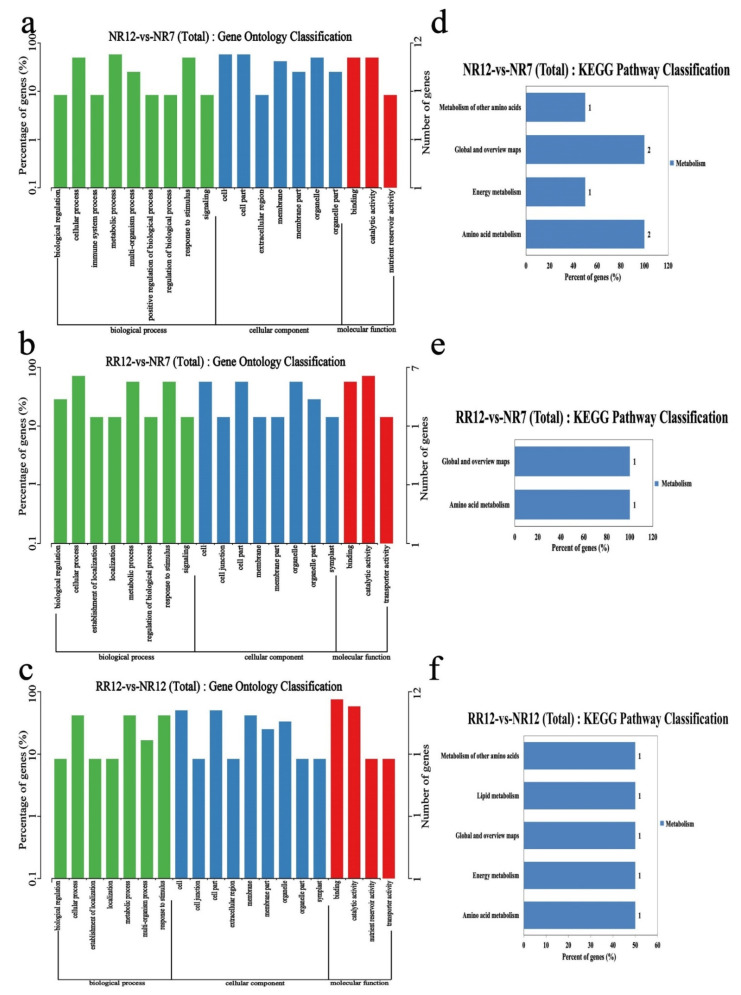
The functional analysis results of host genes of differentially expressed circRNAs. GO enrichment (**a**–**c**) and KEGG pathway (**d**–**f**) in NR12_vs_NR7, RR12_vs_NR7, RR12_vs_NR12.

**Figure 9 genes-13-01547-f009:**
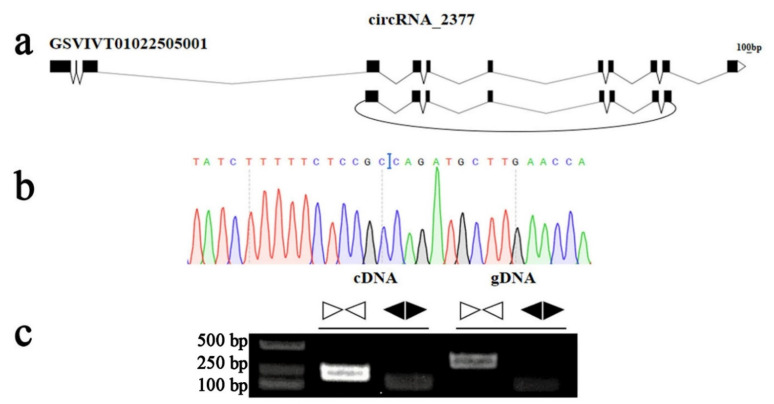
PCR test was conducted to validate the stable expression of grape circRNA_2377. (**a**) According to the genomic loci, circRNA_2377 derived from GSVIVT01022505001 gene. (**b**) Sanger sequencing further confirmed back-splicing site (blue line). (**c**) The agarose gel electrophoresis image showed the expected size of PCR product by divergent primers in cDNA and gDNA. ▹◃ represents convergent primers, ◂▸ represents divergent primers.

**Figure 10 genes-13-01547-f010:**
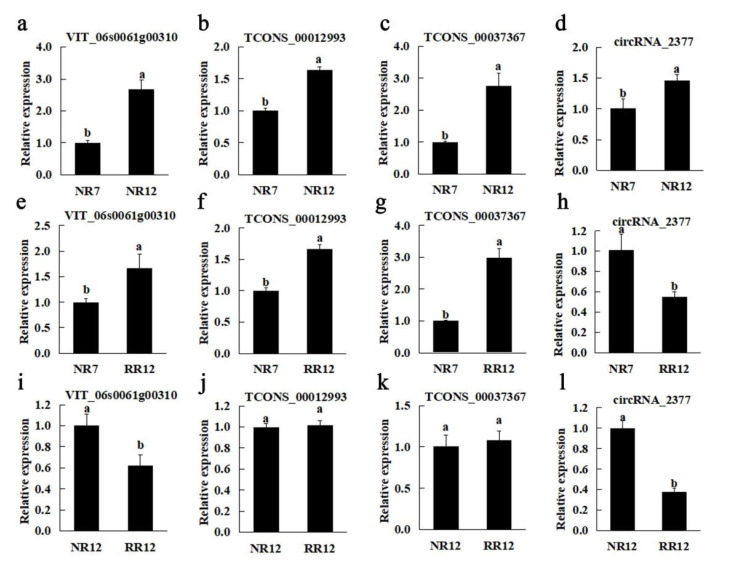
Relative expression analyses of mRNAs, lncRNAs, and circRNAs in the roots of three comparison groups. NR12_vs_NR7 (**a**–**d**), RR12_vs_NR7 (**e**–**h**), RR12_vs_NR12 (**i**–**l**). Significant difference at *p* < 0.05 is indicated by different letters above the columns.

**Figure 11 genes-13-01547-f011:**
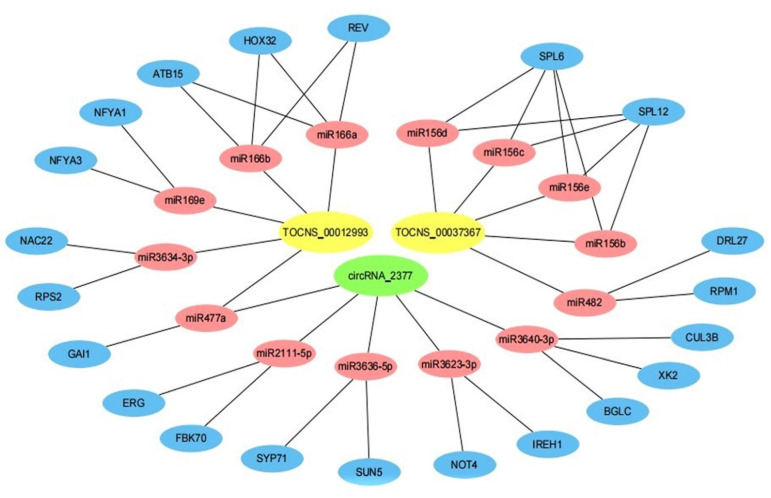
Prediction of interaction network among lncRNAs, circRNAs, miRNAs, and target genes. Yellow, green, red, and blue represent lncRNA, circRNA, miRNA, and target genes, respectively.

**Table 1 genes-13-01547-t001:** The statistical qualification of high throughput sequencing.

Sample	Raw Reads	Raw Bases	Clean Reads	Clean Bases	Valid Bases	Q30	GC
NR12_1	50.65 M	7.60 G	50.23 M	7.27 G	95.70%	93.66%	47.69%
NR12_2	39.34 M	5.90 G	39.03 M	5.67 G	96.12%	93.64%	47.31%
NR12_3	39.31 M	5.90 G	39.00 M	5.66 G	96.07%	93.74%	46.50%
NR7_1	50.00 M	7.50 G	49.58 M	7.12 G	95.00%	94.25%	46.90%
NR7_2	48.08 M	7.21 G	47.72 M	6.89 G	95.54%	94.28%	46.57%
NR7_3	37.01 M	5.55 G	36.74 M	5.35 G	96.38%	93.62%	46.85%
RR12_1	42.49 M	6.37 G	42.13 M	6.13 G	96.22%	93.56%	46.97%
RR12_2	60.20 M	9.03 G	59.79 M	8.69 G	96.20%	93.85%	47.20%
RR12_3	47.77 M	7.16 G	47.37 M	6.84 G	95.40%	93.76%	47.64%

**Table 2 genes-13-01547-t002:** Statistical table of sequence prediction information.

Term	All	NR12_vs_NR7-Diff	RR12_vs_NR7-Diff	RR12_vs_NR12-Diff
mRNA	26,588	2320	1864	2440
lncRNA	1971	176	173	137
circRNA	2615	16	17	9

## Data Availability

The databases used in this study as follows, NCBI database: https://www.ncbi.nlm.nih.gov/sra/PRJNA808789 accessed on 20 February 2022.

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
