# Peer review of "Integrative Investigation of Root-Related mRNAs, lncRNAs and circRNAs of “Muscat Hamburg” (Vitis vinifera L.) Grapevine in Response to Root Restriction through Transcriptomic Analyses"

_genes, 2022, doi:10.3390/genes13091547_

Round 1

Reviewer 1 Report

In this manuscript, the authors investigated transcriptome of grapevine under root restriction. Along with mRNA, they also studied non-coding RNAs, such as lncRNA and circRNA, as these could potentially have a role in the root development.

I have the following comments and suggestions for this manuscript:

- Overall, the text needs to go through extensive spelling and grammar checks. To list a few issues in the text: line 123 "atages", line 259 "through limits the roots", line 362 "adpter", line 380 "flodchange", line 390 "analysised", line 398 "convergent and divergent primers was", line 420 "We totally identified". 

- Please spell out all abbreviations in the first usage. Two examples: RR and FPKM

- In the introduction, it is mentioned that "A total of 26588 mRNAs, 1971 lncRNAs and 2615 circRNAs were identified" and in the conclusion it is referred as "We totally identified two lncRNA and one circRNA involved in the regulation of root development" Does this mean 2 lncRNA and 1 circRNA were confirmed in additional experiments? Should the conclusion be: these lncRNA and circRNA "might be involved in root development", as the functions are still not known?

-In line 77, it is mentioned that the root morphology was significantly changed, does this refer to visual observation? If that is the case, I suggest mentioning "visually".

- For Figures 1 and 2, where NR and RR plants are given, it is hard for the reader to compare the plants in separate figures. Is it possible to pair up NR-RR for a paired comparison?

- I suggest Figure 3 given as a boxplot and include dots of individual measurements. I was surprised to see p<0.05 in 12th day with that large difference of weight of young roots.

- In lines 123-124, it is mentioned that NR7, NR12 and RR12 were selected for sequencing. Could you provide more rationale why these time points were selected? Particularly why was RR7 not picked since it is the time that the difference was observed visually? Also, NR7/RR7 pair could be used in comparisons. Or why was NR/RR9 not selected where the statistical significance was first observed? 

-Could you expand on the difference of the patterns in Figure 4 d-f? Why would we have a different grouping in circRNA compared to mRNA/lncRNA? It is interesting that for circRNA, NR7 and RR12 are closer to each other compared to RR12 (even though with less variance)

-In Figure5, should the y-axis be "counts", instead of the "numbers"?

-I found Figures 7-9 hard to go through. Is it possible to have a summary of these results with most significantly enriched pathways? In the figures, % of gene overlap is given, are these all statistically significant results with p<0.05?

-In line 364, a link to the "reference genome of the experimental specie" should be given, where the data can be reached.

-In line 375, the references to bowtie2 and eXpress are missing, please add.

-In line 382, what is the p-value threshold that is used for significant enrichment?

Reviewer 2 Report

The manuscript entitled “Integrative investigation of root-related mRNAs, lncRNAs and circRNAs of ‘Muscat Hamburg’ (Vitis vinifera L.) grapevine in response to root restriction through transcriptomic analyses “ by Liu et al., analyzes the effect of growing grapes in a root restriction (RR) Condition versus no restriction condition (NR). The authors analyzed the transcriptional changes in mRNAs, lncRNAs and circRNAs. They found that RR increased the root system weight and likely lateral root number. They also made a comparison between NR and RR conditions and NR at two different stages and identified a large number of lncRNAs and circRNAs.

In general, the article is quite interesting to understand root development and also designing new approaches to cultivate crops and increase productivity. However, there are several issues that the author should resolve and improve.

They should better explain why they selected the NR7 and NR12 and RR12 time points to analyze the transcriptome. This is not clear at all and should be justified.  In fact, if they want to compare the effect of NR versus RR they should only use a similar stage of root development. However, if they want to analyze developmental changes they should use NR or RR at different stages. This point, what they want to demonstrate, is not clear.

They mentioned, and analyzed by qRT-PCR, candidate genes related to root development. They should mention the function of these genes and why are related to root development. This is also mentioned in the discussion, and similarly, it should be indicated the name/function of these genes.

According to their hypothesis, if a ceRNA is upregulated in one condition, and these ceRNAs targets a miRNA, it should be expected that the miRNA-target level changes in response to such conditions. This has not been demonstrated by the authors and should be done., at least using the RNAseq data that they present.

They claim several times that RR induces lateral root formation. This has not been shown. They only showed an increase in root weight. If they want to claim a LR increase, they should quantify the number.

The discussion should be improved. In many parts sounds more as a results section than a discussion. Aldo the organization should be improved. They discussed about the function of several miRNAs in different parts of the discussion and they might consider grouping them.

Minor points:

-          In figure 10, it would help the audience to indicate in 10a the localization of the divergent and convergent primers. In fig 10c, there is some amplification in the gDNA with the divergent primers. Why? It should be mentioned and explained.

-          In the Discussion, they said: “During root development, RR treatment had a positive effect on root weight, and the young roots were significantly higher than NR treatment (Figure 3).” What do the authors mean by “were significantly higher”?

-          They should review the grammar and spelling throughout the document since there are many typos and missense phrases.

Round 2

Reviewer 1 Report

I would like to thank the authors for responding to my comments in the revised version. I have 3 more comments/questions for the manuscript:

1. There is one particular section that needs clarification. Please see the lines 103-117 in the clean version of the manuscript:

- First sentence is missing a verb.

- There is a repetition of "further" in the beginning.

- "....RR9-RR12 was heavier than NR9-RR12": Is that supposed to be RR9-12 roots were heavier than NR9-12?

-  "which were in keeping with phenotype of root system": What does this mean?

- Last sentence should be revised to clarify the meaning, particularly "owns to the lateral root morphogenesis on the condition of RR"

2. In addition to the section mentioned above, the manuscript needs to go through another round of language check. Please see below for some issues (please note that this is not an extensive list, the rest of the document also needs to be checked):

- line 50 "RR can also accelerates"

- line 138 "the three independent biological replicates were reliably repeatable ility"

-line 199 incomplete sentence: "To investigate the potential function implication of lncRNAs."

3. The lines 241-243 include "The expression pattern was generally consistent with the results of high-throughput sequencing, indicating that the research method used in this experiment is reliable"

- How was the "general consistency" determined? Please clearly state in the manuscript how this conclusion was obtained. This is also mentioned in line 298.

-Is the consistency through up-regulation level of the RNA checked in NR-RR comparison? How many genes needs to be checked to conclude that the two methods are consistent?

-Is RNA-sequencing the research method that is being referred to? 
